# *Physalis alkekengi* L. Calyx Extract Alleviates Glycolipid Metabolic Disturbance and Inflammation by Modulating Gut Microbiota, Fecal Metabolites, and Glycolipid Metabolism Gene Expression in Obese Mice

**DOI:** 10.3390/nu15112507

**Published:** 2023-05-28

**Authors:** Lin Li, Xiaolong Wang, Ying Zhou, Na Yan, Han Gao, Xiaojie Sun, Chunjing Zhang

**Affiliations:** 1Department of Medical Technology, Qiqihar Medical University, Qiqihar 161006, China; dalin36@163.com (L.L.); s1xiaochong@126.com (X.W.); zhouyou15146109614@163.com (Y.Z.); gaohan041423@qmu.edu.cn (H.G.); sunxj97@163.com (X.S.); 2Department of Basic Medical Science, Qiqihar Medical University, Qiqihar 161006, China; 18345068065@163.com

**Keywords:** *Physalis alkekengi* L. calyx, obesity, glycolipid metabolism, gut microbiota, inflammation

## Abstract

*Physalis alkekengi* L. calyx (PC) extract can relieve insulin resistance and has glycemic and anti-inflammatory effects; however, the potential mechanisms related to gut microbiota and metabolites remain elusive. This study aimed to understand how PC regulates gut microbiota and metabolites to exert anti-obesogenic effects and relieve insulin resistance. In this study, a high-fat high-fructose (HFHF)-diet-induced obesity C57BL/6J male mice model with glycolipid metabolism dysfunction was established, which was supplemented with the aqueous extract of PC daily for 10 weeks. The results showed that the PC supplementation could effectively cure the abnormal lipid metabolism and maintain glucose metabolism homeostasis by regulating the expression of adipose metabolic genes and glucose metabolism genes in the liver, thereby effectively alleviating the inflammatory response. PC treatment also increased the contents of fecal short-chain fatty acids (SCFAs), especially butyric acid. PC extract could restore the HFHF-disrupted diversity of gut microbiota by significantly increasing the relative abundance of *Lactobacillus* and decreasing those of *Romboutsia*, *Candidatus_Saccharimonas*, and *Clostridium_sensu_stricto_1*. The negative effects of the HFHF diet were ameliorated by PC by regulating multiple metabolic pathways, such as lipid metabolism (linoleic acid metabolism, alpha-linolenic acid metabolism, and sphingolipid metabolism) and amino acid metabolism (histidine and tryptophan metabolism). Correlation analysis showed that among the obesity parameters, gut microbiota and metabolites are directly and closely related. To sum up, this study suggested that PC treatment exhibited therapeutic effects by regulating the gut microbiota, fecal metabolites, and gene expression in the liver to improve glucose metabolism, modulate adiposity, and reduce inflammation.

## 1. Introduction

For the past few years, the dietary structure of people has changed significantly with the improvement of the economic level to a diet rich in processed foods and beverages that are high in energy, fat, and sugar. The changes in dietary structure have resulted in the rapid growth of the global overweight and obesity populations [1,2]. The long-term intake of a high-fat and high-fructose diet (HFHF) leads to an excessive intake of high-energy-density fat, which causes the imbalance of energy metabolism and induces the disorder of lipid and sugar metabolism [3], thereby leading to obesity and other related chronic metabolic diseases, such as metabolic syndrome; non-alcoholic fatty liver, cardiovascular, and cerebrovascular diseases; and diabetes [4,5,6].

The liver, as the main organ for lipid metabolism, is an important target organ for obesity. The intake of an HFHF diet increases the burden of liver glycolipid metabolism, causes lipid peroxidation, and affects the normal physiological function of liver and belly cells [7,8]. Moreover, the long-term intake of the HFHF diet can also increase the total liver weight and liver lipid contents, mainly including total cholesterol (TC), triglyceride (TG), and free fatty acid (FFA) contents [9]. This results in the excessive accumulation of fats in the liver and leads to glucose metabolic disorder due to high plasma glucose and insulin levels, resulting in insulin resistance (IR) [10,11].

In recent years, the gut microbiota has attracted the attention of researchers working on metabolic diseases worldwide [12]. The regulation of the gut microbiota can efficaciously improve and prevent the occurrence and development of obesity, IR, and chronic inflammation [13]. Some basic research results show that obesity can increase intestinal permeability, the relative lack of diversity in the gut microbiota [14], a decrease in Bacteroidetes, and an increase in the number of Firmicutes in proportion to a healthy gut microbiota [15,16]. Short-chain fatty acids (SCFAs) are the most abundant microbial metabolite complex derived from carbohydrates and plant polysaccharides in the gut, which can provide energy for microorganisms and the human body [17,18]. Some prebiotics have been used in clinical trials to regulate intestinal microbiota composition and the content of SCFAs in order to reduce obesity and regulate the blood lipid level of the subjects and achieved a good effect [19,20].

Oral administration is the most important mode to administer traditional Chinese medicine (TCM). After entering the intestine, TCMs are in close contact with intestinal microbiota and mainly manifested by regulating the structure of the gut microbiota or the metabolism of active components in TCM, which is an important mechanism for the pharmacological effects of TCM. Huang-Lian-Jie-Du-Decoction (HLJDD) could ameliorate hyperglycemia and restore the dysbiosis of gut microbiota [21]. Corn silk extract also had a decreasing lipoglycemic effect by modulating the fecal metabolites and gut microbiota among hypercholesterolemic mice, which were induced by a high-fat diet [22]. Qingzhuan tea extract supplementation in the diet can improve the symptoms of metabolic syndrome in mice induced by a high-fat diet by regulating the gut microbial structure [20]. All these studies showed the good preventive effect of TCM extracts on ameliorating hyperglycemia and hyperlipemia by regulating the gut microbial structure or metabolism.

*Physalis alkekengi* L. is commonly known as brocade lantern, red girl, etc., and is a perennial plant of the Solanaceae family of herbs. It is a kind of herb resource used both as food and medicine, and diabetes patients often use its calyx as tea to regulate blood glucose in China [23]. Moreover, both its resting calyx and the fruit within are edible, and their extracts have been investigated for their anti-inflammatory glycemic effects, insulin resistance relief effects, and antioxidant effects [24,25,26,27]. It has been confirmed that the substances exerting these pharmacological effects are polysaccharides, sterols, and flavones in fruit and calyx [28]. Although the medicinal value of calyx extract in regulating glycolipid metabolism has been reported, the specific mechanism of lowering glycolipid has not been systematically studied. Research on the effects of supplement PC extract in the diet on the structure of gut microbiota and related metabolite profiles is still scant.

The present study aimed to investigate the effects of PC extract on regulating gene expression in glycolipid metabolism as well as the gut microbiota composition and metabolites in feces in HFHF-diet-fed mice. This study might enhance the understanding of how PC exerts anti-obesogenic and insulin-resistance-relieving effects by modulating gut microbiota, fecal metabolites, and glycolipid metabolism gene expression.

## 2. Materials and Methods

### 2.1. Materials and Reagents

For the preparation of the PC extract, the calyx of the *P. alkekengi* was naturally dried and crushed through a 2 mm sieve followed by extraction with distilled water (90 °C) for 15 min (1:15 *w*/*v*), gently stirring, and extraction twice [29]. The two water extracts were then mixed. In order to facilitate storage, the extracted mixture was centrifuged for 5 min at 5000 rpm, and the supernatant was taken and concentrated under reduced pressure for 1 h at 65 °C. Finally, the powder obtained by vacuum freeze-drying was stored at −20 °C for further use. The contents of polysaccharides, total phenols, and total flavonoids in the water extract were 45.11 mg/g, 1.15 mg/g, and 1.72 mg/g, respectively. The polysaccharides estimation was performed using the phenol-sulfuric acid method, total flavonoids analysis by the AlCl_3_ spectrophotometric method, and total phenols measurement using the Folin–Ciocalteu method. The metabolites of the PC extract were analyzed by Beijing BioMarker Technology Co., Ltd. (Beijing, China). In total, we detected 1381 metabolites from freeze-drying the extract, and most of them belong to fatty acyls, Carboxylic acids and derivatives, Organooxygen compounds, Prenol lipids and Steroids, and steroid derivatives (Appendix A).

### 2.2. Animals and Experimental Design

Six-week-old male C57BL/6 mice, weighing 20.0 ± 1.0 g, were purchased from Liaoning Changsheng biotechnology Co., Ltd. (Liaoning, China). All the animal experiments and procedures were approved by the Ethics Committee of the Laboratory Animal Center, Qiqihar Medical University, Qiqihar, China (approval no. QMU-AECC-2021-172). All mice were fed by the experimental animal center of Qiqihar Medical University (SYXK (HEI) 2016-001). After one-week acclimatization, a total of 28 mice were divided into four groups randomly (*n* = 7 per group): in the NC group, the mice were fed with a standard diet, which contain 10% fat by energy; in the HFHF group, the mice were fed with high-fat feed (60% fat by energy) and a high-fructose diet (drinking water supplemented with 30% fructose); and in the PCL and PCH groups, the mice were fed with a high-fat high-fructose diet and administered with PC 300 mg/kg/day (PCL group) and 600 mg/kg/day (PCH group) by gavage every day (the experimental protocol is detailed in Figure 1A). The experiment lasted for 10 weeks, and the animal weights and food intakes were recorded weekly during the study. At the end of the study, the fresh fecal samples were collected and stored at −80 °C for subsequent analyses of gut microbiota and metabolites as well as SCFAs. After fasting overnight, the mice were sacrificed, and their liver, kidney, epididymis, and subcutaneous fatty tissues were collected, weighed, and stored at −80 °C. The mice serum samples were obtained by centrifugation (1200× *g*, 15 min) and stored at −80 °C for further use.

### 2.3. Biochemical Analysis

The serum triglyceride (TC), total cholesterol (TG), non-esterified fatty acid (NEFA), low-density lipoprotein cholesterol (LDL-C), and high-density lipoprotein cholesterol (HDL-C) levels as well as enzymes levels, including alanine aminotransferase (ALT) and aspartate aminotransferase (AST), were measured using an auto-biochemistry analyzer (Hitachi Ltd., Tokyo, Japan). The levels of inflammatory factor tumor necrosis factor α (TNF-a), interleukin-6 (IL-6), interleukin-1β (IL-1β), monocyte chemotactic protein-1 (MCP-1), lipopolysaccharide (LPS), and serum fasting insulin (FINS) were detected by using an ELISA kit (Nanjing Jiancheng Institute of Bioengineering, Nanjing, China).

The mice were fasted overnight for 12 h prior to the oral glucose tolerance test (OGTT). A 2 g/kg glucose solution was intragastrically administered, and the tail venous blood was taken after 0, 30, 60, and 120 min of the intragastric administration, using a blood glucose meter (Jiangsu Yuyue Medical Equipment & Supply Co., Ltd., Danyang, China). A line graph was drawn using the obtained blood glucose concentration as the vertical coordinate and the corresponding measurement time as the horizontal coordinate, and the area under the curve (AUC) was calculated using the trapezoid rule [30]. The homeostasis model assessment parameter of insulin resistance (HOMA-IR) was calculated using the following formula: HOMA-IR = FINS (mU/L) × FBG (mmol/L)/22.5, HOMA insulin sensitivity (HOMA-IS) index = 1/(FBG × FINS).

### 2.4. Histopathological Analysis

The mice liver and adipose tissues were dissected, extracted, and then fixed with 4% paraformaldehyde. After fixation, the tissue sections were stained with hematoxylin and eosin (H&E), and the structure of liver and adipose tissues was observed and analyzed under an optical microscope [31].

### 2.5. Quantitative Real-Time PCR

Total RNA was extracted from the liver tissues using a Biozol reagent (Invitrogen, Carlsbad, CA, USA), as described previously [32]. The concentration of extracted RNA was determined using a NanoDrop spectrophotometer (BioTeke, Beijing, China). The RNA was reverse-transcribed into cDNA using a reverse transcriptase kit (TransGen Biotech, Beijing, China) following the manufacturer’s instructions. The relative expression levels of genes involved in glycolipid metabolism as well as inflammatory factors (IL-6, IL-β, and TNF-a) and MCP-1 in the liver were measured. The PCR amplification was performed with *Trans Start*^®^ Top Green qPCR SuperMix (TransGen Biotech, Beijing, China) using the ABI 7500 Fast Dx PCR instrument (Applied Biosystems, Foster City, CA, USA). The relative expression levels of the quantitative genes were identified using the 2^−ΔΔCt^ method, with the β-actin set as the reference gene and the NC group as the control. The primer sequences are listed in Appendix A.

### 2.6. Analysis of Short-Chain Fatty Acids

The determination of SCFAs (acetic acid, propionic acid, butyric acid, isobutyric acid, and valeric acid) was performed using gas chromatography (GC) by Shanghai Personalbio Technology Co., Ltd. (Shanghai, China). First, the contents of fecal samples were collected, samples were thawed on ice, and 30 mg of the sample was placed in a 2 mL glass centrifuge tube. Then, 900 μL 0.5% phosphoric acid was added and resuspended, and the mixture was shaken for 2 min. After centrifugation at 14,000× *g* for 10 min, 800 μL of the supernatant was taken, and the same amount of ethyl acetate was added for extraction, shaken, mixed for 22 min, and centrifuged at 14,000× *g* for 10 min. The upper organic phase of 600 μL was mixed with 4-methylvaleric acid (final concentration 500 μM) as an internal standard and then added into the injection bottle for GC-MS detection. The injection volume was 1 μL, and the split ratio was 10:1. MSD ChemStation software was used to extract chromatographic peak areas and retention times. The standard curve was drawn to calculate the SCFAs in the sample [32].

### 2.7. Analysis of Gut Microbiota

Total genomic DNA was extracted from the fecal samples using the MOBIO PowerSoil^®^ DNA Isolation Kit (MOBIO Laboratories, Inc., Carlsbad, CA, USA), followed by the amplification and sequencing of hypervariable V3-V4 region in the 16S rRNA gene using the Illumina Novaseq 6000 platform by Beijing Bio Marker Technology Co., Ltd. (Beijing, China). The detailed analysis methods are described in a previous study [32]. USEARCH software was used for the clustering of sequence reads with a 97.0% similarity level, and OTUs were obtained [33]. Then, the composition of gut microbiota in each sample at each level (phylum, class, order, family, genus, and species) was determined, and the species abundances at different classification levels were identified using the QIIME software. The α and β diversities were analyzed to reveal the diversity and structure of gut microbiota. β diversity analyses included non-metric multidimensional scaling (NMDS) and principal coordinate analysis (PCoA) based on the Bray–Curtis algorithm; ANOSIM analysis was used to test for significant differences in the clusters among the groups [34]. LEfSe (linear discriminant analysis (LDA) effect size) was used to find biomarkers with statistical differences between different groups.

### 2.8. Untargeted Metabolomics Analysis

Fecal metabolome analysis was performed by Beijing BioMarker Technology Co., Ltd. (Beijing, China). The analysis mainly included thawing, grinding, extraction, vacuum drying of fecal samples, etc. The detailed steps are following the reference and our previous study [35,36] and in Appendix A.

### 2.9. Statistical Analysis

Among different groups, statistical significance was set at *p* < 0.05 and determined with one-way ANOVA by using the Tukey–Kramer post hoc test. The results were expressed as mean ± SD. The correlations found among variables were identified by Pearson’s product-moment correlation coefficient. Statistical tests were performed using R software (Version 4.2.1) for Windows.

## 3. Results

### 3.1. PC Prevented HFHF-Induced Obesity in Mice

During the experimental period (Figure 1A), the mice groups showed changes in their weights after 10 weeks (Figure 1B). The initial weights were similar among the four groups. The final body weights of mice in the NC, HFHF, PCL, and PCH groups were 30.58 ± 1.16 g, 40.92 ± 1.71 g, 37.88 ± 1.11 g, and 34.64 ± 1.33 g, respectively. As compared to the NC, the mice weight increased more rapidly in the HFHF group, and the weight of the mice in the PCL and PCH groups decreased significantly by 7.43% and 15.35%, respectively, as compared to those in the HFHF group (Figure 1C, *p* < 0.05). There were no differences in the daily food intake by mice among the four groups, showing that the suppressed body weight effect of PC did not come from the reduction in food consumption (Figure 1D, *p* > 0.05). Consistently, the PC administration reduced the weights of liver and subcutaneous tissues as compared to those in the HFHF group. Moreover, the high-dose administration of PC also significantly reduced the weight of epididymal tissue (Figure 1E, *p* < 0.01); however, the weight of kidneys did not change among the four groups (*p* > 0.05). H&E staining showed that the PC treatment significantly reduced the cell diameter of adipocytes (Figure 1F).

**Figure 1 nutrients-15-02507-f001:**
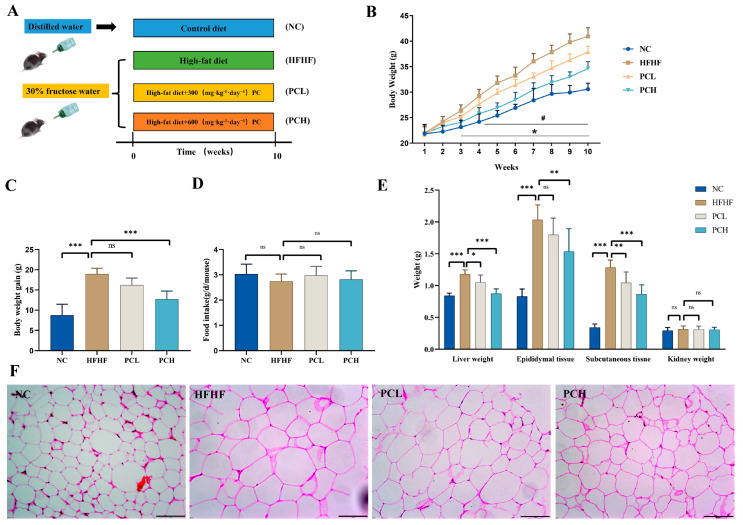
(**A**) The experimental scheme, *n* = 7 in each group; and effects of PC supplementation on the (**B**) body weight. * *p* < 0.05, PCH compared with HFHF; ^#^
*p* < 0.05, PCL compared with HFHF. (**C**) Body weight gain, (**D**) food intake, (**E**) organ weights, and (**F**) histology analysis of adipose tissue. The length of the black line in the figure represents 50 μm. Data are expressed as means ± SD (*n* = 7). * represents *p* < 0.05, ** represents *p* < 0.01, *** represents *p* < 0.001, and ns represents not significant.

### 3.2. PC Improved Glucose Homeostasis and Alleviated HFHF-Induced Insulin Resistance

After the 10-week experimental period, OGTT was used to evaluate glucose homeostasis (Figure 2). The results indicated that the blood glucose level was higher in the HFHF group as compared to that in the NC group, while PC supplementation significantly decreased the blood glucose level in HFHF-fed mice (*p* < 0.05). The OGTT showed that the oral glucose tolerance of the HFHF group mice was lower than that of the NC group mice, with a significantly increased AUC value (*p* < 0.01). Consistently, the PC administration significantly prevented the HFHF-induced impairment of OGTT. Obesity, caused by a high-fat and high-fructose diet, is also closely related to insulin resistance. As compared to the NC group, the FINS level increased significantly in the HFHF group, which was indicated by a significant increase in the HOMA-IR index and a significant decrease in the HOMA-IS index. The PC supplementation for 10 weeks significantly prevented these adverse changes, leading to a decrease in the FINS level and HOMA-IR index and a significant increase in HOMA-IS index (*p* < 0.05) in the PCL and PCH groups as compared to those in the HFHF group.

### 3.3. PC Attenuated the HFHF-Induced Dyslipidemia

Obesity is usually accompanied by fat accumulation and elevated blood lipids. As shown in Figure 3A–E, the serum levels of TG, TC, NEFA, and LDL-C were significantly increased in HFHF mice as compared to those in the NC group. The PC supplementation reversed these indices. Surprisingly, there was no significant difference in the serum levels of HDL-C in mice among the four groups. Additionally, the HFHF-fed mice were characterized by higher levels of TG, TC, and NEFA in the liver, which were also significantly reversed by PC supplementation (Figure 3F–H). These data suggested that the PC could protect mice against hyperlipidemia induced by HFHF.

As compared to the NC group (Figure 3I,J), the serum AST and ALT levels increased significantly in the HFHF group (*p* < 0.01). The high-dose PC supplements effectively prevented HFHF-induced liver damage by ameliorating the serum AST and ALT levels as compared to those in the HFHF group (*p* < 0.01); however, we did not observe significant changes in ALT levels between HFHF and PCL groups. H&E staining also showed that the PC treatment reduced the ballooning degeneration in the liver significantly (Figure 3K). These results suggested that PC supplementation could effectively suppress HFHF-induced liver lipotoxicity.

### 3.4. PC Modified the Expression Profiles of Genes Involved in Glycolipid Metabolism in Livers

In order to further explore the effects of PC on glycolipid metabolism in the liver were investigated. As compared to NC, the expression levels of lipid synthesis genes (Figure 4A), including *FAS*, *ACC1*, *SCD1*, *SREBP-1c,* and *ChREBPα* increased significantly in the HFHF group; these increased expression levels were significantly downregulated by PCH treatment as compared to those in the HFHF group (*p* < 0.05). The expression levels of fatty acid oxidation genes, such as *Pparα*, *Cpt1α,* and *Acox1* significantly increased in the PCH group as compared to those in the HFHF group (Figure 4B, *p* < 0.05).

In order to reveal the mechanism of lowering the blood glucose level and improving glucose homeostasis by PC, the expression levels of two gluconeogenesis-related genes (*G6Pase* and *PEPCK*) in the liver were determined (Figure 4C). The expression levels of *G6Pase* and *PEPCK* in the mice liver in the HFHF group were significantly upregulated compared with the NC group. The PC treatment significantly downregulated the expression of these two genes.

### 3.5. PC Inhibited the HFHF-Induced Secretion of Proinflammatory Cytokines

To evaluate the anti-inflammation effects of PC, the serum levels of proinflammatory cytokines, including TNF-α, IL-1β, IL-6, MCP-1, and LPS, were examined in mice. As shown in Figure 5A,B, the serum levels of TNF-α, IL-1β, and MCP-1 increased significantly in the HFHF group as compared to the NC group (*p* < 0.05), while that of IL-6 showed no significant changes (*p* > 0.05). Under the HFHF diet, the PC supplement could significantly reduce the serum levels of TNF-α, IL-1β, and MCP-1 as compared to HFHF-induced mice, and the effects of the PCH group were better than those of the PCL group. Moreover, HFHF significantly increased the mRNA expression levels of TNF-α, IL-1β, IL-6, and MCP-1 in liver tissues as compared to those in the NC group (Figure 5C). The PCH treatment significantly reduced the expression levels of these proinflammatory cytokines (TNF-α, IL-1β, and MCP-1) in the liver as compared to those in the HFHF group. There were no significant differences in the expression levels of TNF-α and IL-6 in mice liver between PCL and HFHF groups (*p* > 0.05). The PC supplementation significantly reduced the serum level of LPS in HFD-induced mice in a dose-dependent manner (*p* < 0.01, Figure 5D).

### 3.6. PC Altered the Fecal SCFAs Composition

As shown in Figure 6, the concentrations of SCFAs in feces were determined. In the HFHF group, the contents of acetate, butyrate, and total SCFAs decreased by 17.17%, 44.41%, and 16.38%, respectively, as compared to those in the NC group; this decrease was reversed by the PCH supplementation. Moreover, the butyrate acid content increased significantly in the PCL group as compared to that in the HFHF group. This suggests that PC supplementation could increase the butyrate content. However, there were no differences in the contents of propionate, valeric acid, and isobutyric in the four groups.

### 3.7. PC Modulated Gut Microbiota Composition at Different Taxonomic Levels

The V3-V4 hypervariable region of the 16S rRNA gene was sequenced on an Illumina MiSeq platform to assess the effects of PC on the gut microbiota composition. A total of 1,650,700 clean reads were detected after quality filtering in the 28 samples (*n* = 7 per group) and obtained 58,954 clean reads per sample on average. The Chao1, Shannon, and Simpson indices were measured to evaluate the effect of PC on gut microbiota community richness and microbial evenness (Figure 7A). As compared to the NC group, the Chao1 index and Simpson index decreased significantly in the HFHF group, and the Shannon index decreased with no statistical significance (*p* > 0.05); this decrease was reversed by PC supplementation in the PCH group (*p* < 0.05). This indicated that PC could restore the HFHF-induced low diversity of gut microbiota.

Beta diversity was assessed using PCoA and NMDS on the Bray–Curtis algorithm to show the effects of PC on the community composition of gut microbiota (Figure 7B,C). PCoA realizes the classification of multiple samples and shows the differences in species diversity among different samples. The closer the samples are on the coordinate map, the greater the similarity between them. Principal coordinates were used for the three-dimensional expansion (Figure 7B), where the contribution rates of PC1, PC2, and PC3 were 32.47%, 15.05%, and 8.79%, respectively. The samples in the NC and PCH groups were mainly concentrated in the plane formed by PC1 and PC3, while the samples in the HFHF and PCL groups were mainly concentrated in the PC2, indicating that PCH could partly restore the health status of HFHF-disrupted gut microbiota. The NMDS of the microbial composition also revealed that the PCH treatment groups had different microbial community structures as compared to those in the HFHF groups (Figure 7C).

To further explore the effects of PC on the composition of gut microbiota, we identified the microbiota relative abundance at the phylum, family, and genus levels to demonstrate the essential changes induced by HFHF and PC. At the phylum level (Figure 7D), Firmicutes, Bacteroidetes, Actinobacteria, Patescibacteria, Proteobacteria, and Desulfobacterota were the top six most abundant phyla and formed 97% of total gut bacteria in the four groups. The results showed that the gut microbiota in the HFHF group consists of much more Firmicute with less Bacteroidetes as compared to those in the NC group, leading to an increase in the Firmicute/Bacteroidetes ratio (F/B ratio); however, the high-dose PC supplementation reversed these changes (*p* < 0.05, Figure 7E–G). The high-does PC treatment also prevented an HFHF-induced increase in the Firmicute/Bacteroidetes ratio (F/B ratio) (Figure 7G), which is a hallmark of obesity and a common indicator for the imbalance of gut microbiota. Moreover, as compared to the NC group (Figure 7H), the relative abundance of Patescibacteria was much higher in the HFHF group and significantly lower in the PCL and PCH groups (Figure 7H). At the family level, the top three abundant families were *Bacilli*, *Clostridia*, and *Bacteroidia* in the four groups. The relative abundances of these three bacterial families significantly changed after PC treatment as compared to those in the HFHF group (Appendix A). PCH could effectively reverse the HFHF-induced decrease in the relative abundance of *Bacteroidetes*; similarly, PCL treatment could also increase the relative abundance of *Bacteroidetes*, but the difference was not significant. Both the PCL and PCH treatments decreased the relative abundance of *Saccharimonadia* as compared to that in the HFHF group. PCH also significantly decreased the relative abundance of *Clostridia*. These results indicated that low-dose PC treatment had little effect on the composition of gut microbiota.

Furthermore, the top 15 abundant bacterial genera are shown in Figure 8A,B. The HFHF feeding increased the relative abundances of *Ligilactobacillus*, *Romboutsia*, *Candidatus_Saccharimonas*, *Clostridium_sensu_stricto_1*, and *Monoglobus* as compared with the NC group. This increase was reversed by the PCH treatment (*p* < 0.05). As compared to the HFHF group, the genus *unclassified_Muribaculaceae* was more enriched in the PCH groups, and the genus *Lactobacillus* was more enriched in the NC and PCH groups. LEfSe analysis showed PCH group enrichment in *unclassified_Muribaculaceae,* whereas the HFHF group was enriched in *Clostridium_sensu_stricto_1* (LDA scores > 4.5, Figure 8C).

The altered glycolipid metabolism and inflammation index might reflect the functions of the gut microbiota. Therefore, the functional correlations were explored between significantly differential bacteria at the genus level and obesity parameters associated with glucolipid metabolism and inflammation (Figure 8D). The results showed that *Romboutsia*, *Candidatus_Saccharimonas*, *Clostridium_sensu_stricto_1*, and *Monoglobus* were significantly positively correlated with the parameters of glycolipid metabolism and inflammation and negatively correlated with acetic acid (AA) and butyric acid (BA), which belonged to SCFAs. Lactobacillus showed a negative correlation with the parameters of glycolipid metabolism and inflammation except for FINS and serum TG as well as a significant positive correlation with BA.

### 3.8. PC Altered the Gut Metabolic Profile in Mice

Untargeted fecal metabolomic analysis based on the LC-QTOF platform and qualitative and quantitative metabolomic analysis were performed on 28 samples. A total of 10,230 peaks were detected in the positive ionization mode, among which 2417 metabolites were noted, and 10,291 peaks were detected in the negative ionization mode, among which 2025 metabolites were noted. To investigate the fecal metabolite differences among NC, HFHF, PCL, and PCH groups in mice, PCA and OPLS-DA were performed (Figure 9). The PCA results showed that the NC group was clearly distinguished from the other three groups for both positive (POS) and negative (NEG) modes (Figure 9A). To further reflect the differences between treatments (Figure 8B,C), the OPLS-DA results of the permutation test validated the model and revealed that fecal metabolites significantly differed between NC and HFHF mice in both POS and NEG modes (Figure 9B), and the results demonstrated that the fecal metabolites were altered in the PCH group compared to the HFHF mice (Figure 9C). It also showed different changes between PCL and HFHF groups (Appendix A). The result suggests that PC supplementation could regulate the metabolic profiles of the mice.

The volcano plot was applied to determine the overall trend of differences in metabolite contents between the two groups. The statistical significance of the differences and metabolites with VIP > 1, *p* < 0.05, and fold change (FC) ≥ 2 was significantly altered by PC (Figure 10A–D). As compared to the NC group, in the positive ionization mode, 299 metabolites were upregulated and 162 metabolites were downregulated in the HFHF group, while in the negative ionization mode, 88 metabolites were upregulated and 186 metabolites were downregulated. In the PCH group, 337 metabolites were upregulated and 296 metabolites were downregulated in the positive ionization mode, while in the negative ionization mode, 74 metabolites were upregulated and 256 metabolites were downregulated as compared to the HFHF group. There were also significant differences in metabolites between the PCL and HFHF groups (Appendix A). Based on the volcanic map results, hierarchical clustering heat maps are further drawn for the screened differential metabolites (Figure 10E,F).

The Human Metabolome Database (HMDB) was used to annotate and classify the differential metabolites. (Figure 10G, Appendix A). The results indicated that compared to the NC group, most of the HFHF-induced differential metabolites belonged to lipids and lipid-like molecules, organic acids and derivatives, organoheterocyclic compounds, benzenoids, and organic nitrogen compounds. The high-dose PC supplementation regulated most of these metabolites, including lipids and lipid-like molecules, organoheterocyclic compounds, organic acids and derivatives, benzenoids, phenylpropanoids and polyketides, and organic oxygen compounds. Most subclasses belonged to gycerophosphates glycerophosphoethanolamines, amino acids, peptides and their analogs, lineolic acids and derivatives, bile acids, alcohols and derivatives, fatty acids and conjugates, diradvloivcerols, carbohydrates, carbohvdratecoiuoates, etc. (Appendix A). In order to further explore the metabolic pathways associated with these changed metabolites by PC supplementation, the KEGG database was used for the pathway enrichment analysis. The cluster profiler hypergeometric test was used to enrich and analyze the annotation results of differential metabolites, and a bubble map was drawn. The redder the dot color, the more significant the enrichment, and the dot size represents the number of differentiated enriched metabolites. The top 20 significant pathways are shown in Figure 10H and Appendix A. The top 20 metabolic pathways, which were regulated after PCH treatment, are also shown. The metabolic pathways with the highest enrichment mainly included linoleic acid metabolism, one carbon pool by folate, alpha-linolenic acid metabolism, sphingolipid metabolism, histidine metabolism, folate biosynthesis, glycerophospholipid metabolism, tryptophan metabolism, purine metabolism, and steroid hormone biosynthesis. This showed that these pathways were mainly involved in lipid and amino acid metabolism pathways. These results suggested that the PC supplementation could alter the fecal metabolic profile of glycolipid and amino acid metabolism in mice.

Gut microbiota and metabolites are closely correlated. These correlations were explored using Spearman’s correlation analysis and screened using volcano plots (Appendix A). The result showed that *Candidatus_Saccharimonas*, *Romboutsia*, *Clostridium_sensu_stricto_1*, and *Monoglobus* were positively correlated with pregnenolone, pregnenolone sulfate, 19-hydroxyandrost-4-ene-3,17-dione, and glycerophosphocholine and negatively correlated with GlcCer (d18:1/16:0).

### 3.9. Correlations Network among Gut Microbiota, Differential Metabolites in Feces, Glycolipid Metabolism Parameters in Serum, and Inflammatory Factor in Mice

There were multiple correlations among the serum indices, gut microbiota, and metabolites (Figure 11). All serum glycolipid metabolism indices (TC, TG, FBG, and FINS) were strongly negatively correlated with SCFAs (jasmonic acid, 2-hydroxybutyric acid, succinic acid, L-(+)-lactic acid, and methylmalonic acid), and the SCFAs in feces (AA and BA) were also significantly negatively correlated with these glycolipid metabolic indices (*p* < 0.05). This further confirmed that hyperlipidemia and hyperglycemia could decrease SCFA production, which could be reversed by PC supplementation. Prasteron-sulfate, GlcCer (d18:1/16:0), 9-hydroxylinoleic_acid, 13(S)-HpoDE, 9(S)-HpODE, and 9,10-DiHOME, belonging to glycolipid metabolites, were strongly negatively correlated with glycolipid metabolism indices and immune factors in serum (TC, TG, FBG, and FINS). Among them, GlcCer (d18:1/16:0) was strongly negatively correlated with *Candidatus_Saccharimonas*, *Clostridium_sensu_stricto_1*, and *Monoglobus*, which were positively correlated with parameters of glycolipid metabolic indices in serum. These results indicated that the HFHF-induced disturbance of serum glucose and lipid metabolism is closely related to gut microbiota and metabolites.

## 4. Discussion

Although it has been reported that PC has the functions of lowering blood pressure as well as anti-inflammatory, anti-tumor, and hypoglycemic activities [28], the effects of PC as a dietary supplement on gut microbiota, inflammation, and NAFLD have not been investigated. The present study showed that PC was protective against HFHF-induced glucose and lipid metabolism disorder, and its effect was associated with regulating the expression levels of liver glucolipid metabolism genes, systemic inflammation, gut microbiota, and fecal metabolites.

Hyperlipidemia is the most common risk factor for cardiovascular and cerebrovascular diseases and is closely related to the intake of a high-fat diet [37]. Animal experiments showed that the intake of a high-fat diet could significantly increase the levels of plasma TC, TG, and LDL-C and significantly reduce that of HDL-C in rats [38,39]. The current study found that PC significantly decreased fat accumulation and hyperlipidemia in HFHF-fed mice. This may be because PC contains flavonoids and polyphenols. Studies have shown that supplement dietary flavonoids and polyphenols can alleviate HFD-induced obesity, thereby reducing the atherosclerosis index [40,41]. A long-term HFHF diet can further cause liver damage and lipid deposition in the body, thereby causing an irreversible cycle in the body. Moreover, abnormal liver lipid metabolism is related to insulin resistance. Multiple studies have shown that excessive sugar intake leads to impaired glucose tolerance and insulin resistance [42,43]; this was consistent with the results of the current study. Wu et al. showed that the PC water extract could improve the tolerance of mice to sucrose and glucose [44]. The current study also confirmed that PC supplementation significantly inhibited the symptoms of impaired glucose tolerance and insulin resistance in HFHF-fed mice. Therefore, this study confirmed that PC had a preventive and therapeutic effect on the disorder of glucose and lipid metabolism.

De novo lipid (DNL) synthesis refers to the conversion of dietary glucose into fatty acids, which further synthesize TG [45]. The liver and adipose tissues are the main organs responsible for DNL synthesis [46]. Due to a series of enzymatic reactions and cytokines involved in regulating lipid anabolic metabolism, liver DNL synthesis is more efficient than that on adipose tissue. Moreover, some anti-obesity and hypoglycemic drug candidates could prevent lipid and blood glucose levels by regulating liver glucose and lipid metabolism. Fatty acid synthase (*FAS*), acetyl-coA carboxylase 1 (*ACC1*), and stearoyl CoA desaturase (*SCD*) are the key enzymes in natural fatty acid synthesis [47]. A high-fat diet could significantly upregulate the transcription levels of *FAS* and *ACC1* in the liver and increase the lipid synthesis load in the liver as well as significantly increase the levels of TG and LDL-C [31,48]. The current study showed that high-dose PC supplementation could significantly downregulate the expression levels of *FAS*, *ACC1*, and *SCD1* in the liver. As compared to the HFHF model group, the PC supplementation also significantly downregulated the expression levels of the *SREBP-1c* gene. *SREBP-1c* mainly regulates genes related to triglyceride and fatty acid synthesis, such as *ACC* and *SCD1* [49]. Recent studies have shown that *SREBP-1c* can also inhibit the transcription of insulin receptor substrate-2 (*IRS-2*), thereby exacerbating insulin resistance; this suggested that it might have a negative regulatory effect on insulin signaling [50,51]. *ChREBPα* is a transcription factor in the glucose signaling pathway, which plays an important role in regulating glucose metabolism [52], fat metabolism, and fat deposition in mammals [53]. It acts synergistically with *SREBP-1c* to regulate the expression of glycolysis and fatty acid synthetase genes. In this study, the PC supplementation significantly reduced the *ChREBPα* gene expression, inhibited lipid synthesis, and regulated abnormal glucose metabolism.

Fatty acid oxidation metabolism in the liver is another important pathway affecting the formation of a fatty liver. *PPARα* plays a key role in regulating lipid metabolism, glucose homeostasis, and the anti-inflammatory response, and the upregulation of *PPARα* expression at the mRNA level could increase fatty acid catabolism and decrease fat mass, thus decreasing TG and LDL-C levels in the liver [32,54]. *Cpt1α* was the target gene of the *PPARα*-mediated fatty acid oxidation pathway, and the *Acox1* gene is a key enzyme in the peroxisome oxidation pathway. A high-fat diet can reduce the expressions of *PPARα*, *Cpt1α,* and *Acox1* genes in the liver of model group mice, indicating that the lipid oxidative metabolism pathway of non-alcoholic fatty liver mice was blocked by a high-fat diet [55]. PC supplementation improves the expression of those three genes. The above research indicated that PC could inhibit the occurrence of lipid metabolism disorder and adipose tissue deposition in HFHF mice by regulating gene expression involved in lipid synthesis and oxidation metabolism.

To verify the effects of PC on glucose homeostasis in HFHF-fed mice and reveal the mechanism of PC on improving insulin resistance, the expression levels of two gluconeogen-related genes, phosphoenolpyruvate carboxykinase (*PEPCK*) and glucose-6-phosphatase (*G6Pase*), in mice liver were determined in this study. The deficiency of these major enzymes in T2DM resulted in the dysregulation of glucose homeostasis [56]. The current results indicated that the mRNA expression levels of *G6Pase* and *PEPCK* in mice liver were significantly downregulated in the PC treatment, which might play an important role in inhibiting liver gluconeogenesis.

Chronic low-grade inflammation is an important characteristic of high-fat-diet-induced obesity [57]. Inflammatory cytokines are also the key factors of insulin resistance. Studies showed that obesity-induced proinflammatory cytokines were closely related to insulin resistance [58,59]. However, in insulin-resistant individuals, a large number of proinflammatory cytokines and inflammatory mediators, especially TNF-α, MCP-1, and IL-6, were upregulated [60]. The current research illustrated for the first time that PC could significantly inhibit HFHF-induced inflammatory response, which was manifested in the reduction of inflammatory factors (TNF-α, IL-1β, and MCP-1) in serum and liver tissues.

However, when gut microbiota is disturbed, the abundance of harmful microorganisms increases, disrupting the intestinal barrier function, increasing plasma endotoxin levels, and exacerbating systemic inflammation in the host. The gut microbiota disturbance is closely related to the occurrence and development of NAFLD, insulin resistance, and inflammation [61]. In contrast, SCFAs are important metabolites, which play an important role in regulating host metabolism. Emerging evidence has shown that gut microbiota compositional alteration and short-chain fatty acids (SCFAs) reduction were observed both in fructose-fed mice [62] and high-fat-fed mice [63]. The PC supplementation partially restored the HFHF-induced decrease in SCFAs, such as butyrate and acetate contents. Butyrate can especially significantly relieve steatohepatitis by regulating the gut microbiota and intestinal barrier function, thereby reducing inflammation and oxidative damage in the liver, and it also has a good effect on blood glucose and energy balance [64].

The addition of PC not only ameliorates the high-fat-diet-induced disorder of glucolipid lipid metabolism but also plays a certain role in regulating the structure and function of gut microbiota. The PC supplementation restored microbial diversity, which was reduced by the HFHF diet. Numerous studies have confirmed that the abundance of Firmicutes increases, while that of Bacteroides decreases in obese individuals [63,65], suggesting that the gut microbiota of these two phyla play an important role in developing obesity. The gut microbiota of HFHF-fed mice was characterized by a lower abundance of Bacteroidetes and an increased abundance of Firmicutes and F/B ratios. PC extract was rich in polysaccharides. Wu et al. also showed that polysaccharides from *P*. *alkekengi* could reverse the Bacteroidetes/Firmicutes ratio and reduce lipopolysaccharide generation and inflammation-related bacteria in insulin-resistant mice [66]. These results suggested that PC supplementation could reduce HFHF-induced gut ecological disorder and further improve liver fat accumulation.

Previous studies have demonstrated a causal relationship between gut microbiota regulation and glycolipid metabolism homeostasis. Our study found that PC supplementation could significantly reduce the relative abundance of harmful gut microbiota. *Romboutsia*, *Clostridium_sensu_stricto_1*, and *Candidatus_Saccharimonas* are generally perceived as pathogenic bacteria and interpreted as indicators of less healthy gut microbiota; moreover, they are correlated with obesity, inflammation, dyslipidemia, and necrotic enteritis [32,67,68]. The Spearman’s correlation analysis also showed that *Romboutsia*, *Clostridium_sensu_stricto_1*, and *Candidatus_Saccharimonas* were positively correlated with glucolipid metabolism and inflammatory factors in this study, and the PC supplementation significantly restored the relative abundances of those microbes. Wang et al. [69]. reported that the abundance of *Monoglobus* increased with disease activity in rheumatoid arthritis (RA) and was positively correlated with the IL-10, TNF-α, and IFN-γ levels. The current study demonstrated that the relative abundance of *Monoglobus* significantly decreased in PC treatment and was positively correlated with the level of proinflammatory factors. A previous study showed that *Monoglobus* might have a protective effect on chronic kidney disease [70]. Therefore, the specific effects of *Monoglobus* on the host under the disease state requires further clarification. In addition, the PC supplementation also significantly increased the abundance of *Lactobacillus*. According to previous studies, a high-fat diet could significantly decrease the abundance of *Lactobacillus*, which was closely related to the level of lipid metabolism. The correlation analysis of gut microbiota and related physiological and biochemical indicators in this study indicated that the relative abundance of *Lactobacillus* was negatively correlated to related lipid metabolic disorder indicators, and it also inhibits chronic inflammation and the exacerbation of hyperglycemia. To a certain extent, these results indicated that PC could improve the richness of probiotics in the gut microbiota of obese mice.

The alteration of gut microbiota composition is always accompanied by significant functional alteration and unavoidably leads to alteration in the metabolites in the gut. Fecal metabolites are the key linkage between gut microbiota and host health. In order to explore the important metabolites involved in the beneficial effects of PC extract, the changes in metabolite profile and metabolic pathways were analyzed using fecal samples. The PLS-DA analysis and volcano plot suggested that the PC supplementation could regulate the metabolic profiles of the mice. Using KEGG enrichment analysis, it was found that the metabolic pathways with the highest enrichment in PC groups were mainly involved in lipid metabolism (linoleic acid metabolism, alpha-linolenic acid metabolism, sphingolipid metabolism, glycerophospholipid metabolism, and steroid hormone biosynthesis) and amino acid metabolism pathways (histidine metabolism and tryptophan metabolism). Studies have shown that histidine can promote lipid reduction, increase insulin sensitivity, and reduce the levels of systemic inflammatory markers in plasma [71], while tryptophan can protect mammalian cells against oxidative stress agents. Sphingolipid plays an important biological role; its metabolism was affected by a high-fat diet, and the gut microbiota can produce sphingolipids to improve resistance to stress [72].

The PC treatment could also regulate glycerophospholipid metabolism and steroid hormone biosynthesis. Glycerophospholipids are the main components of biological membranes and have significant implications for cellular functions; moreover, they are also closely related to the immune system [73]. As a precursor of highly unsaturated fatty acids (HUFAS), alpha-linolenic acid is an essential polyunsaturated fatty acid having various biological functions. The signaling molecules generated after the metabolism of alpha-linolenic acid can regulate the growth of cells. However, the lack of alpha-linolenic acid in the body leads to a disorder of lipid metabolism, a decline in immunity, atherosclerosis, and other diseases [74]. Linoleic acid can also be substituted by intestinal microbiota into co-yoked linoleic acid, which plays a role in improving glycolipid metabolism [75]. Studies have shown that obesity can lead to decreased levels of linoleic and linolenic acid, and a high-fat diet can cause oxidative stress in the gut, leading to over-oxygenation of these two fatty acids [76]. In this study, the contents of α-linolenic acid and linoleic acid in fecal metabolites were regulated after PC intervention, which might have importance for better understanding the potential mechanism of PC intervention in the disorder of glycolipid metabolism.

Recently, metabolomic changes in fecal samples are related to the gut microbiota [77], especially in the occurrence and development of diseases, such as obesity, diabetes, anxiety, etc. Although this study could not determine the causal correlation between gut microbiota and metabolites, the correlation analysis suggested a certain correlation between them. At the same time, we found that serum blood indices were also correlated with fecal metabolites. These results indicated that there was an interaction among “serum indices-gut microbiota-fecal metabolites”. In order to clarify the correlation between the three, future studies should consider fecal microbiota transplantation, inoculation tests of related strains, and the upregulation and downregulation of key metabolites.

## 5. Conclusions

In conclusion, this study concluded that the dietary supplementation of PC could improve the altered host metabolic homeostasis and play a role in beneficial activities against systemic inflammation. The PC treatment not only significantly altered the composition of gut microbiota but also the fecal metabolites. In subsequent studies, regulating the gut microbiota and their related metabolites can be used as a new entry point to study the mechanism of PC. As a medicine and food item, PC has great potential in the treatment and prevention of glucose and lipid metabolism disorders and their complications.

## Figures and Tables

**Figure 2 nutrients-15-02507-f002:**
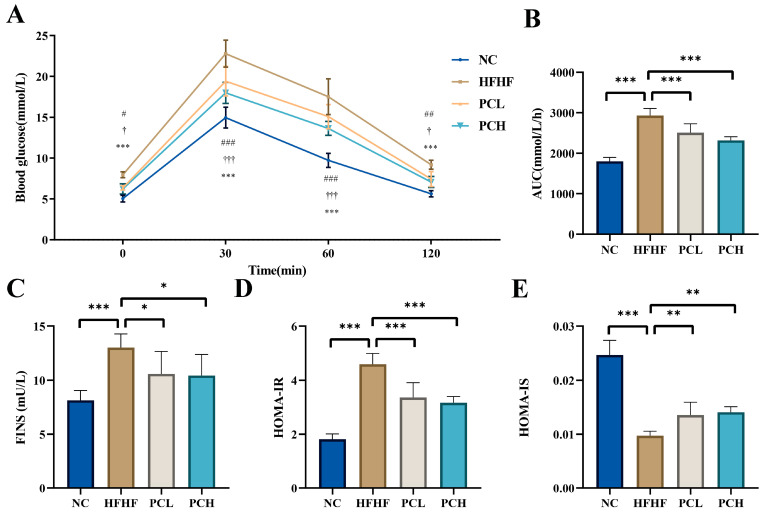
Effects of PC intervention on glucose homeostasis and alleviated insulin resistance in mice. (**A**) The oral glucose tolerance test (OGTT) curve in mice. *** represents *p* < 0.001 when NC group is compared with HFHF group; † represents *p* < 0.05 and ††† represents *p* < 0.001 when PCL group is compared with HFHF group; # represents *p* < 0.05, ## represents *p* < 0.01, and ### represents *p* < 0.001 when PCH group is compared with HFHF group. The area under the curve (AUC) (**B**). Serum fasting insulin (FINS) levels (**C**), and Homeostasis model assessment of insulin resistance (HOMA-IR) (**D**) and insulin sensitive (HOMA-IS) index (**E**). Data are expressed as means ± SD (*n* = 7). * represents *p* < 0.05, ** represents *p* < 0.01, *** represents *p* < 0.001.

**Figure 3 nutrients-15-02507-f003:**
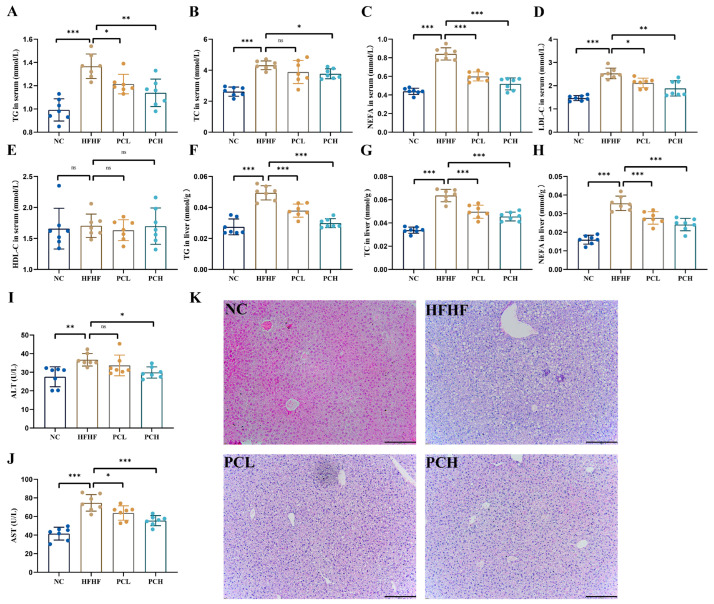
Effects of PC supplementation on the serum and liver (**A** or **F**) triglyceride (TG), (**B** or **G**) total cholesterol (TC), (**C** or **H**) non-esterified fatty acid levels (NEFA), (**D**) low-density lipoprotein cholesterol (LDL-C) in serum, (**E**) high-density lipoprotein cholesterol (LDL-C) in serum, (**I**,**J**) serum alanine aminotransferase (ALT) and aspartate aminotransferase (AST) activities, and (**K**) H&E staining of mice livers. The length of the black line in the figure represents 100 μm. Values are expressed as mean ± SD in each group (*n* = 7). * represents *p* < 0.05, ** represents *p* < 0.01, *** represents *p* < 0.001, and ns represents not significant.

**Figure 4 nutrients-15-02507-f004:**
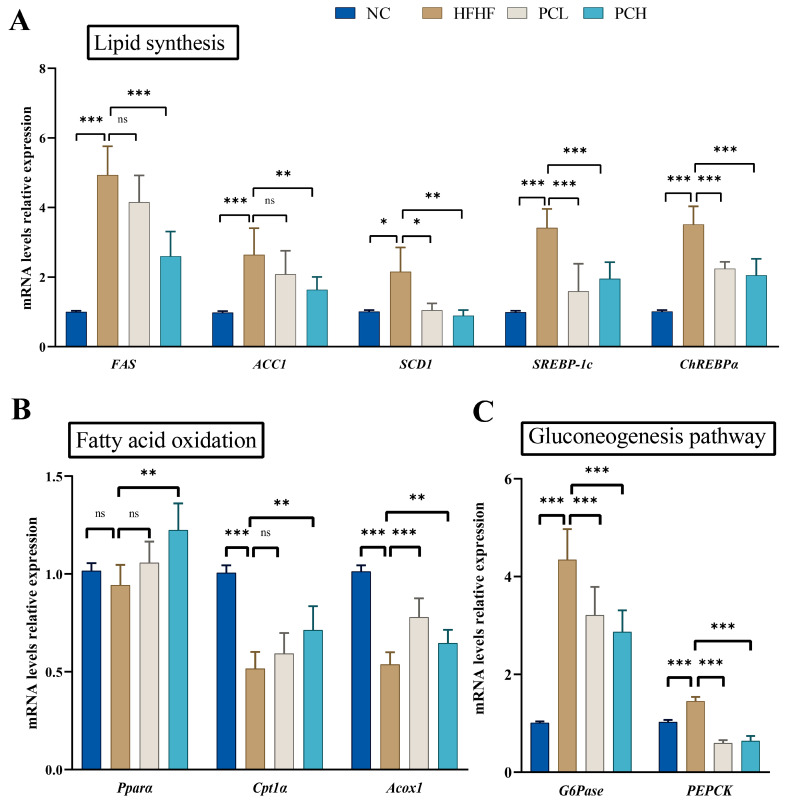
Effects of PC supplementation on the transcription of genes related to glycolipid metabolism in the liver. (**A**) Lipid synthesis, Fas fatty acid synthase (*FAS*); Acetyl-CoA Carboxylase 1 (*ACC1*); stearoyl-CoA desaturase 1(*SCD1*); sterol regulatory element-binding protein-1C (*SREBP-1c*); and Carbohydrate-response element-binding proteins (*ChREBPα*). (**B**) Fatty acid oxidation, peroxisome proliferator-activated receptor α (*Pparα*); carnitine palmitoyltransferase 1a (Cpt1a); and acyl-coenzyme A oxidase 1 (*Acox1*). (**C**) Gluconeogenesis pathway, Glucose-6-phosphatase (*G6Pase*); phosphoenolpyruvate carboxykinase (*PEPCK*). Data are expressed as means ± SD (*n* = 7). * represents *p* < 0.05, ** represents *p* < 0.01, *** represents *p* < 0.001, and ns represents not significant.

**Figure 5 nutrients-15-02507-f005:**
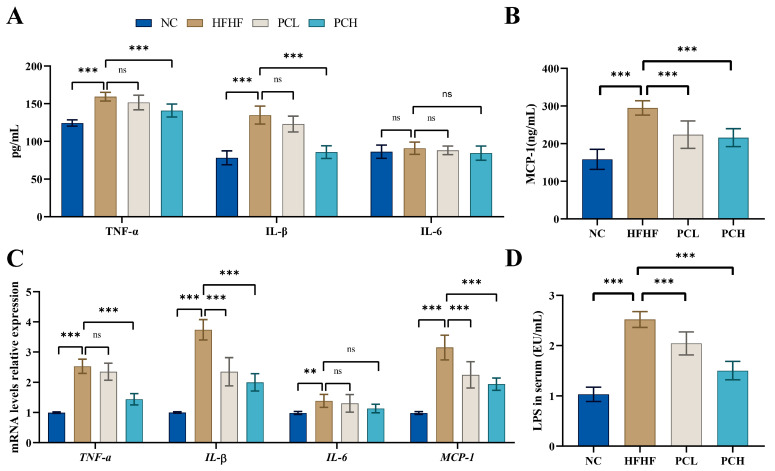
Effect of PC supplementation on inflammation in serum (**A**,**B**) and in hepatic (**C**). Effect of PC on LPS in serum (**D**). Data are expressed as means ± SD (*n* = 7), ** represents *p* < 0.01, *** represents *p* < 0.001, and ns represents not significant.

**Figure 6 nutrients-15-02507-f006:**
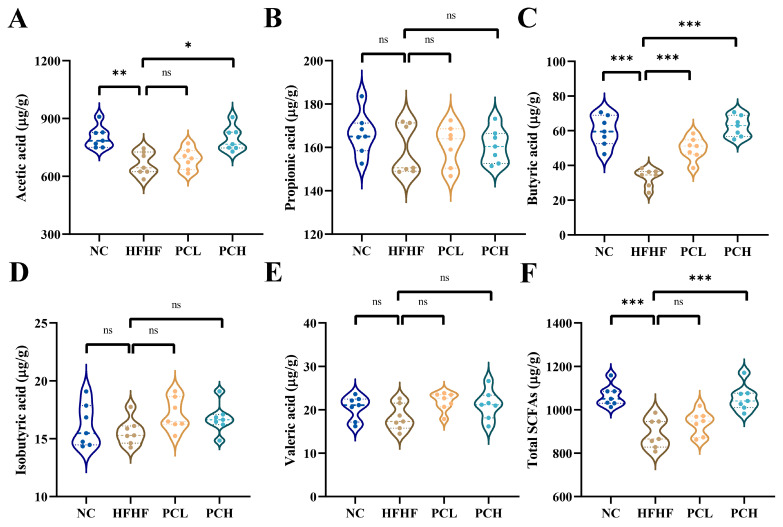
Effects of PC supplementation on SCFAs production in (**A**) acetic acid, (**B**) propionate acid, (**C**) butyrate acid, (**D**) isobutyric acid, (**E**) valerate acid, and (**F**) total SCFAs of fecal contents. Data are expressed as means ± SD (*n* = 7), * represents *p* < 0.05, ** represents *p* < 0.01, *** represents *p* < 0.001, and ns represents not significant.

**Figure 7 nutrients-15-02507-f007:**
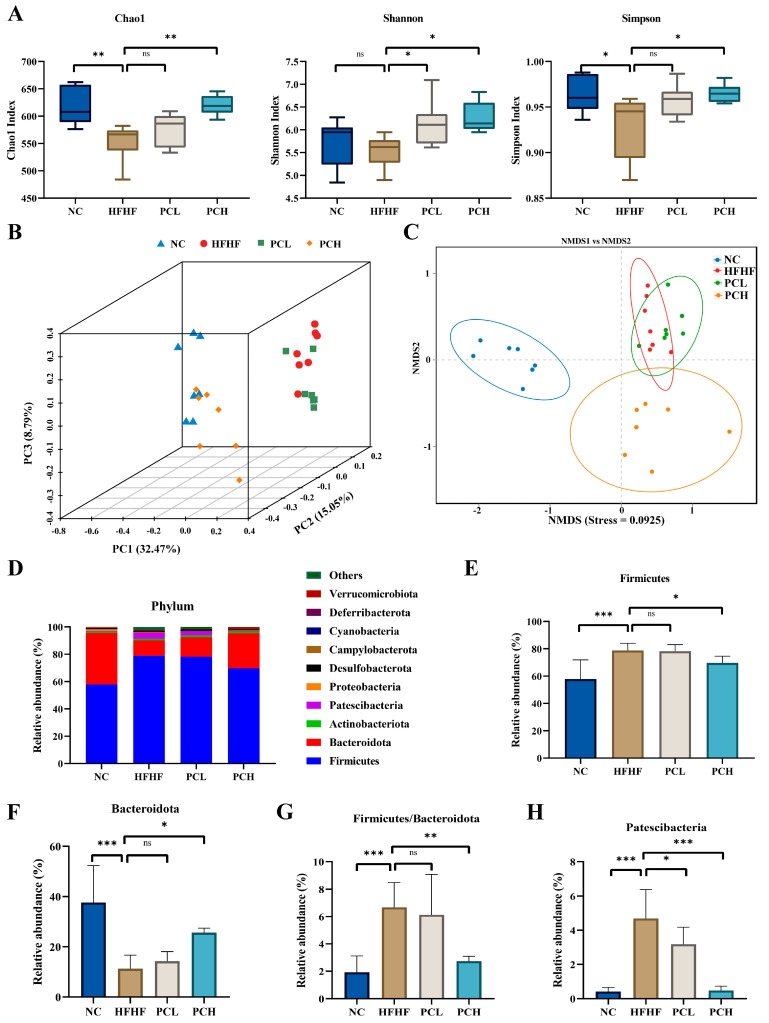
Effect of PC supplementation on composition of the gut microbiota. (**A**) Alpha diversity analysis of Chao1, Shannon, and Simpson index, (**B**) PCoA plot based on Bray—Curtis distance, (**C**) non-metric multidimensional scaling (NMDS), (**D**) bacterial taxonomic profiling in the phylum level, and (**E**–**H**) significantly changes (*p* < 0.05) in the relative abundance of the gut microbiota at phylum taxa level. Data are expressed as mean ± SD, * represents *p* < 0.05, ** represents *p* < 0.01, *** represents *p* < 0.001, and ns represents not significant.

**Figure 8 nutrients-15-02507-f008:**
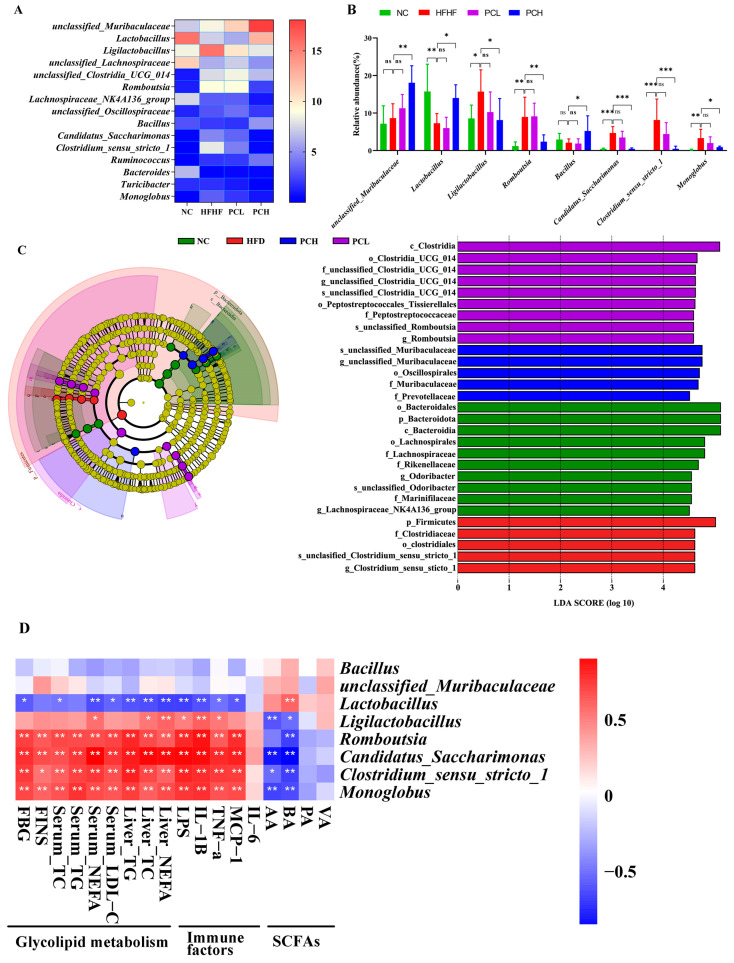
Deep analysis of gut microbiota at genus level in mice. (**A**) The top 15 abundances of the gut microbiota at the genus level in heat map, (**B**) Bacterial genus significantly changed by HFHF, and the treatment of PC among top 15 taxa of the composition of the gut microbiota at genus level. Data are expressed as mean ± SD, * represents *p* < 0.05, ** represents *p* < 0.01, *** represents *p* < 0.001, and ns represents not significant. (**C**) Linear discriminant analysis effect size (LEfSe) analysis of key genera of gut microbiota in mice and the LDA score > 4.5, and (**D**) correlation between the gut microbiota and biochemical indexes at the genus level. * *p* < 0.05, ** *p* < 0.01. AA, acetic acid; BA, butyrate acid; PA, propionate acid; VA, valerate acid.

**Figure 9 nutrients-15-02507-f009:**
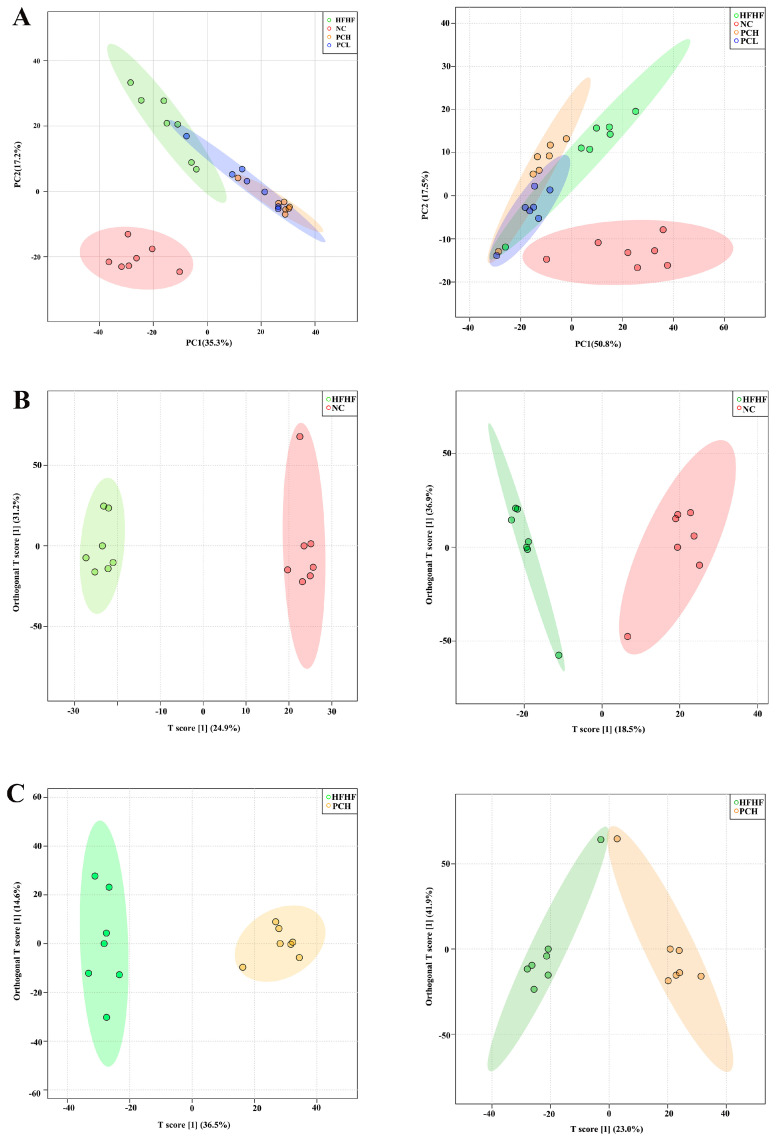
The fecal metabolic profile altered by PC in HFHF–fed mice. (**A**) Score plot of the PCA model in the positive and negative ion modes. (**B**) Score plot of the OPLS–DA model in positive and negative ion modes between NC and HFHF. (**C**) Score plot of the OPLS–DA model in positive and negative ion modes between HFHF and PCH.

**Figure 10 nutrients-15-02507-f010:**
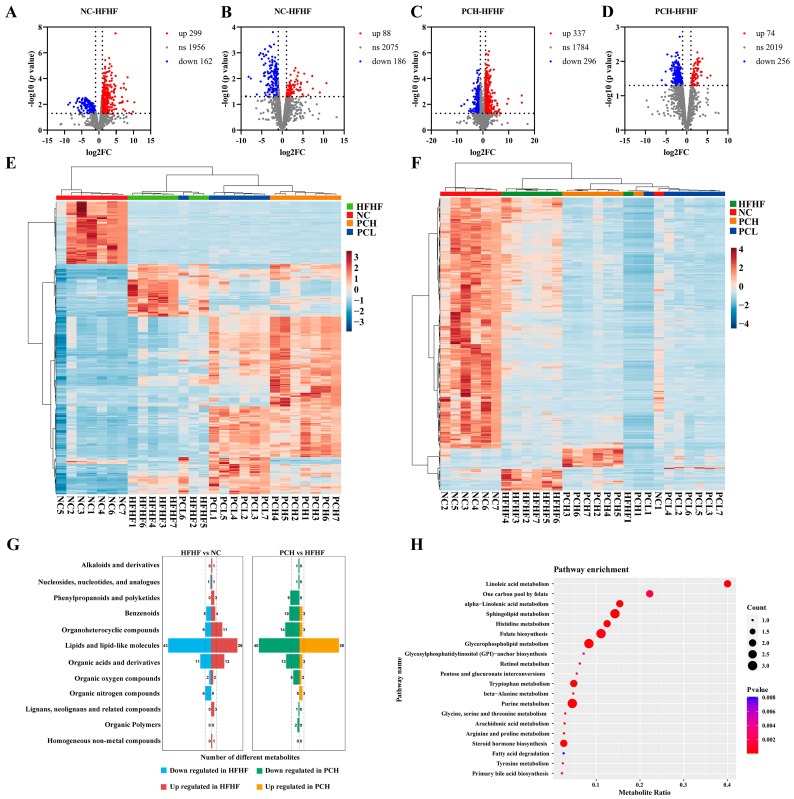
The fecal metabolites altered in mice. Differential metabolites in fecal identification in different groups in (**A**,**C**,**E**) positive and (**B**,**D**,**F**) negative ion modes, and expression of differential metabolites in the two groups was represented by (**A**–**D**) volcano plot and (**E**,**F**) heat maps. (**G**) The number of differential metabolites was annotated and classified based on the HMDB changed by HFHF and the treatment of PCH. (**H**) The enrichment metabolic pathway of fecal altered metabolites by Kyoto encyclopedia of genes and genomes (KEGG) analysis.

**Figure 11 nutrients-15-02507-f011:**
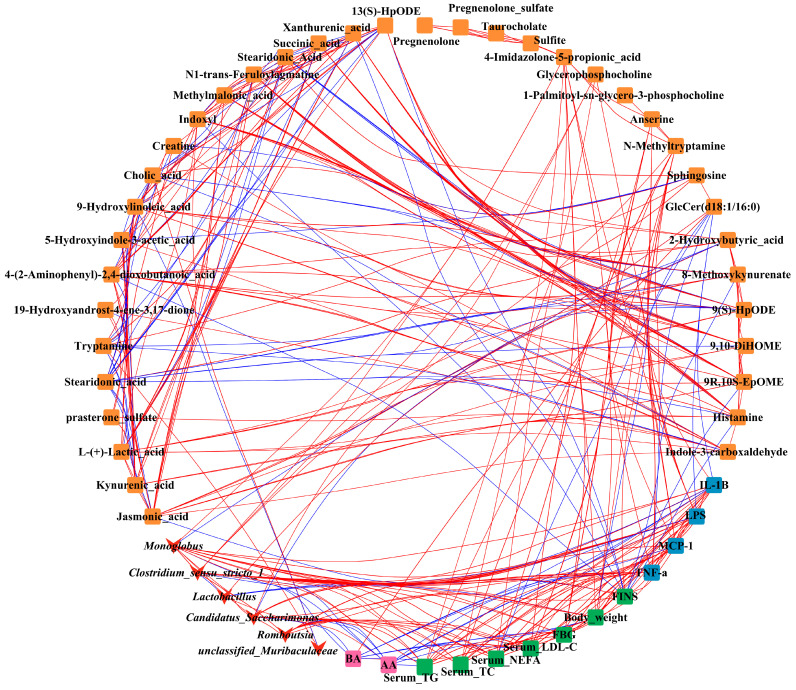
Correlation network analysis showing correlations among significant differences in specific gut microbiota, co-regulated metabolites in feces, and obesity-related biomarkers; the red line represents the positive correlation and the blue line represents negative correlation. |*r*| > 0.8, *p* < 0.05.

## Data Availability

The raw data supporting the conclusions of this article will be made available by the authors, without undue reservation.

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
