# Peer review of "Physalis alkekengi L. Calyx Extract Alleviates Glycolipid Metabolic Disturbance and Inflammation by Modulating Gut Microbiota, Fecal Metabolites, and Glycolipid Metabolism Gene Expression in Obese Mice"

_nutrients, 2023, doi:10.3390/nu15112507_

Round 1

Reviewer 1 Report

Material and Methods

2.1 Materials and reagent 

How do you know that the extraction with hot distilled water didn't degradate the chemical compounds present in the extract?

I would like the reason why you decided to use this extraction method without using a cold extraction. 

Why did not analyze the extract and characterized it with a metabolic profile?

3 Results

Figure 1 C, D, E  

Can you please put stars on the images to enhance the statistical significance between groups? I think that for the reader could be helpful to understand how much effective is your treatment. 

Figure A-J 

Also in this case I ask for a more direct interpretation of the datas, please add significativity on the histograms if present. 

Line 419 it also had (It capital letter please).

Please try to improve the contrast and the saturation of the colour of the F images because considering the violet and the white colours is hard to see clearly the cells. 

Figure 2 

A, B, C, D, E, F, G H

Figure K 

It is clear the effect of your extract from the images however the quality images are low and I would like to ask also for a magnification to appreciate bettere the details. 

Figure 9 E, F 

Please improve the quality of the heat maps, the volcano plot (is very hard to understand the result and to read)instead for me are good and easy to read them. 

Reviewer 2 Report

General analysis of the manuscript…

The revised manuscript presents clearly, orderly and extensively the effects of Physalis alkekengi L. Calyx on the metabolic disorders, systemic inflammation and microbiota changes induced by the administration of high fat and high fructose diet in mice.

The study includes biochemical, anatomo-pathological, molecular, taxonomical and statistical deep analysis that permit to support the conclusions presented.

In accordance to the criteria of this reviewer, the manuscript is able to be published in the Nutrients Journal. Only a minor list of editorial suggestions are listed below…

Minor editorial corrections…

Add spaces in line 13, 45, 47, 51, 53, 56, 62, 69, 71, 73, 79, 83, 166, 168, 175, 181, 502.

Redundancies: 64 – 65 – 66: microbiota.

Change words: line 79… China? Line 117… metabolites?

Suggestions about sentences: In line 109 “NC group, in which the mice were fed with a standard diet, containing 10% fat by energy”, may be will result clearer if you write, “NC group, whereas the mice were fed with a standard diet, which contain 10% fat by energy”.

Please, include a list of acronyms before de introduction section. That will make easy the comprehension reading of the manuscript.

In the figures that show microscope pictures (figures 1 and 2), please, eliminate the number related to the scale bar. Include into the legend that bars indicate 100 µm…

No further comments.

Reviewer 3 Report

The study aimed to understand how Physalis alkekengi L . Calyx (PC) regulates gut microbiota and metabolites to exert anti-obesogenic effects and alleviate insulin resistance. The results clearly showed that PC could effectively restore the abnormal glycolipid metabolism in the liver as well as the HFHF-disrupted diversity of gut microbiota and metabolites in feces. Correlation analysis was also performed to show the relationship between the obesity parameters, gut microbiota, and metabolites. The investigation shows that PC has potential in the treatment and prevention of glucose and lipid metabolism disorders. This work is interesting and the manuscript is well written.

I have the following minor concerns:

(1)    Please describe how the levels of polysaccharides, total phenols, and total flavonoids were determined in the water extract (Line 100-101, Page 3)

(2)    How were the oral administration doses of PC extract determined in animals?

(3)    Was only GC or GC-MSused to determine short-chain Fatty acids? Please clarify in Part 2.6

(4)    Please describe how to assess HOMA-IR and HOMA-IS index.

(5)    Describing the effect of OGTT by Table is unusual. It is better to present the data as a curve.

(6)    Statistical analysis should be clarified. For example, what is the meaning of the bars labeled a-d, to which they are compared?

The language of the manuscript could be further polished.

Reviewer 4 Report

In this manuscript authors evaluated the beneficial effects of Physalis alkekengi L. Calyx on metabolic disturbance and inflammation in vivo. These beneficial effects of Physalis alkekengi L. Calyx are assessed by various markers/parameters like biochemical analysis, histopathological analysis, gut microbiota, and metabolomics analysis in C57/BL6J mice. At last, the author draws a conclusion that Physalis alkekengi L. Calyx could prominently inhibit the development of metabolic disturbance and inflammation in obesity mice. The research question is well defined and execution is also good but there are certain queries and suggestion which authors must need to resolve before accepting this manuscript for publication.

Here are some of these:

1. I suggest revise the title to: Physalis alkekengi L. Calyx Extract Alleviate Glycolipid Metabolic Disturbance and Inflammation by Modulating Gut Microbiota, Fecal Metabolites and Glycolipid Metabolism Genes Expression in Obesity Mice

2. In the 2.2. Animals and experimental design: The description of animal treatments (Line 108-114) is very poor and not conducive to reading. Please revise it.

3. In the Table 1: 8.13 ± 0.40v, What does the letter v mean?

4. In the Table 1: HOMA-IR and HOMA-IS calculation formula needs to be provided in the table annotations.

5. In Figure 3: All gene name abbreviations on the horizontal axis need to be italicized.

6. There is no mention of the LPS detection method and its kit used in the Materials and Methods section.

7. In Figure 6F and H: The strain name is not italicized. The same question also has many places in the main text. Please check and modify one by one, such as Line 353, 356.

8. In Figure 6G: I suggest using full letters instead of abbreviations for the F/B marked at the top.

9. In the Discussion: It is necessary to appropriately discuss the relationship between the main active ingredients in Physalis alkekengi L. Calyx and the regulatory activity of metabolic disturbance and inflammation.
